# Simultaneous Colonic Pressure Waves in Children and Young Adults with Gastrointestinal Motility Disorders: Artefact or Colonic Physiology?

**DOI:** 10.3390/jcm12185808

**Published:** 2023-09-06

**Authors:** Richard Leibbrandt, Samuel Nurko, S. Mark Scott, Phil G. Dinning

**Affiliations:** 1College of Science and Engineering, Flinders University, Adelaide 5042, Australia; richard.leibbrandt@flinders.edu.au; 2Center for Motility and Functional Gastrointestinal Disorders, Boston Children’s Hospital, Boston, MA 02115, USA; samuel.nurko@childrens.harvard.edu; 3National Bowel Research Centre and GI Physiology Unit, Blizard Institute, Queen Mary, University of London, London E1 2AT, UK; m.scott@qmul.ac.uk; 4College of Medicine and Public Health, Flinders University, Adelaide 5042, Australia; 5Department of Gastroenterology and Surgery, Flinders Medical Centre, Adelaide 5042, Australia

**Keywords:** colon, manometry, pediatric, constipation, simultaneous pressure waves, artefact

## Abstract

Background: Simultaneous pressure waves (SPW) spanning all recording sites in colonic manometry studies have been described as a potential biomarker of normal gas transit and extrinsic neural reflexes. In pediatric studies utilizing combined antroduodenal and colonic manometry, it was noted that most colonic SPWs appeared to also span all sensors in the gastric and small bowel regions. This suggests that a proportion of colonic SPWs may represent an artefact caused by forces extrinsic to the colon. Our aim was to characterize colonic SPWs and determine how many of these spanned most of the digestive tract. Methods: In 39 combined high-resolution antroduodenal and colonic manometry traces from 27 pediatric patients, we used our purpose-built software to identify all SPWs that spanned either (i) all recording sites in the digestive tract or (ii) those restricted to the colon. Results: A total of 14,565 SPWs were identified (364 ± 316 SPWs/study), with 14,550 (99.9%) spanning the entire antroduodenal and colonic recording sites. Only 15 SPWs (0.1% of the total) were restricted to the colon (all in one recording). Conclusions: Based on these findings, we suggest that, in pediatric studies, SPWs should not form part of any diagnostic criteria, as these events appear to be an artefact caused by factors outside the colon (abdominal strain, body motion).

## 1. Introduction

Colonic motility, responsible for the mixing and transit of intraluminal content (and ultimately defecation), occurs through coordinated contractions and relaxations of the colonic longitudinal and circular muscle layers. The introduction of high-resolution colonic manometry has helped to record and define various motor patterns in healthy adults and identify potential motor abnormalities in both adults and children with constipation [1]. While propagating motor patterns are, and have been, the focus of many publications, attention has also been paid recently to simultaneous pressure waves that span the entire array of sensors in the colon [2,3,4,5]. These motor patterns have been labelled ‘colonic pressurizations’ [5] or ‘simultaneous pressure waves’ (SPWs) [3] and can occur in relatively high numbers with 85 ± 38/7 h recorded in one study [5]. Previously, these motor patterns have predominantly been labelled as an artefact (principally caused by body movement, coughing, sneezing, or laughing). However, Corsetti et al. [5], utilizing abdominal EMG, reported that colonic pressurizations were not associated with any contraction of abdominal muscles. The physiological role of these motor patterns remains uncertain but their frequency increases during a meal [5], and they have been temporally associated with the transit of gas and also high-amplitude propagating contractions [2]. As such, their presence in a manometry study is proposed as a potential biomarker of normal gas transit and an indicator of normal extrinsic neural reflexes [2].

In pediatric patients presenting with severe and refractory gastrointestinal motility disorders, motility throughout the digestive tract can be assessed by performing manometry of the upper (stomach and proximal small bowel) and lower gut (colon) simultaneously. While conducting an analysis of these data for a recent publication [6], a high incidence of SPWs was noted in the colonic manometry recordings. It was also observed that many of these colonic SPWs appeared to span the entire gastric and small bowel recording, suggesting they represented an artefact related to the extrinsic forces applied to the colon (e.g., body movements or abdominal strain) rather than genuine colonic physiology. The aim of this study was to fully characterize colonic SPWs in patients that had undergone combined antroduodenal and colonic manometry and determine the proportion of these events that could be labelled as ‘colonic-only’ motility patterns, versus those that spanned the entire digestive tract (gastric/small bowel/colonic).

## 2. Materials and Methods

Manometry traces were collated from 27 pediatric or former pediatric patients (now aged > 18) who underwent simultaneous high-resolution antroduodenal and colonic manometry recordings at the Boston Children’s Hospital, performed by one of the investigators (SN). The indication for such pan-intestinal manometric studies was to identify whether there was a motility abnormality that could explain symptoms of refractory constipation or pseudo-obstruction. Twelve of these patients underwent 2 separate recordings on 2 consecutive days, with the catheters remaining in place for both recordings. By including these repeat studies, we were, hence, able to analyse 39 manometry recordings. Meals were provided in 25 of the 39 recordings. The protocol for the procedure was approved by the local ethics committee (IRB P00026190) and informed consent for the procedures was given by the responsible caregivers. The present analysis was performed by investigators blinded to the clinical characteristics of the study subjects.

This is a second retrospective study utilizing some of these data. The first, looking at the interactions between colonic high-amplitude propagating contractions and small bowel motility, has already been published [6].

### 2.1. Specific Procedures: Colonic and Antroduodenal Manometry

All patients were admitted to the hospital and received a standardized inpatient bowel clean out with polyethylene glycol. The following day, whilst under anaesthesia, separate high-resolution, solid-state antroduodenal and colonic manometry catheters were placed endoscopically/colonoscopically. 

The antroduodenal catheter was inserted either through the nose, or, if the patients had one, via a gastrostomy, and advanced into the proximal jejunum, as previously described [7,8]. The catheter (Unisensor; Portsmouth, NH, USA) had 36 recording sensors spaced as follows: one sensor at the tip; 5 cm separating the next 3 sensors; and the remaining 32 sensors separated by 2 cm. In all patients, catheter sensors were located in the stomach, duodenum, and proximal jejunum. In 5, there were also sensors located in the oesophagus.

The colonic manometry catheter was placed with the use of a colonoscope and guide wire, as previously described [9]. The catheter (Unisensor) had 36 sensors, each separated by 3 cm. The catheter tip was placed in the cecum or proximal transverse colon where possible. The colonic catheter was securely taped to the patient’s thigh to avoid displacement.

Due to the 3 cm spacing between sensors, the accurate maintenance of a sensor within the anal canal was not possible. Although previous studies have shown an association between colonic SPWs and anal sphincter relaxation [2,3,5], such an assessment did not form part of the analysis in the current study. 

The patients were kept in the hospital and studies were performed 24 h after the administration of anesthesia to allow for the washout of the anesthetic agents and any effects of the endoscopic procedures [8,10]. The placement of the catheters was checked with a plain X-ray before the initiation of data collection.

### 2.2. Antroduodenal Manometry Protocol

The complete study followed a protocol previously described [7,8]. In brief, manometry was initially performed for a maximum 3 h period of fasting. If phase III of the migrating motor complex (MMC) was not observed during the fasting period, an infusion of erythromycin (1 mg/kg) over a 30 min period was given. If this failed to initiate phase III of the MMC, then octreotide was given. Succeeding the fasting period or after a minimum of 30 min (range 30–180 min) following erythromycin/octreotide administration, a meal was given, followed by a minimum of 1 h of postprandial recording. The type and size of the meal were adjusted according to the patient’s age (10 kcal/kg or 400 kcal with >30% lipid content) and was either given by mouth or into the stomach via the gastrostomy stoma, as indicated. 

### 2.3. Colonic Manometry Protocol

Colonic manometry was performed simultaneously with antroduodenal manometry and, therefore, followed the same protocol. In addition, one hour after the meal, bisacodyl (0.25 mg/kg to a maximum of 10 mg) was infused through the central lumen to be released at the tip of the catheter. 

### 2.4. Definitions of Identified Manometric Signals

*Simultaneous pressure waves (SPWs)* (Figure 1A): These were defined as simultaneous pressure waves recorded by all sensors in the colon and rectum. ‘Simultaneous’ was determined if there was no detectable time lag between the upstroke of pressures in each sequential channel. Based upon a recent publication, all SPWs with a pressure-wave duration between 2 and 57 s were included [2]. SPWs were then divided into 2 categories: (i) ‘colonic-only’ SPWs, which were identified only in the colon/rectum and were not seen in the simultaneously acquired gastric/small bowel recordings and (ii) ‘gastric/small bowel/colonic’ SPWs, defined by simultaneous pressures waves spanning all sensors in the colon and in the stomach/small bowel. 

*Interrupted colonic pressurizations* (Figure 1B); In addition to the SPWs described above, it became apparent that some events, while clearly not simultaneous across all channels, did consist of multiple short-extent simultaneous pressure waves, occurring at spatial locations that appeared to ‘advance’ each segment of simultaneous pressure waves aborally. These sequences were reminiscent of the subclass of “interrupted” colonic pressurizations described by Corsetti et al. [5], and, hence, we have used the same terminology in this paper. Interrupted colonic pressurizations were only identified in the colon.

*High amplitude propagating contractions (HAPCs)* (Figure 1C,D) were identified to determine their association (if any) with SPWs. These motor patterns were defined as a series of monophasic pressure peaks occurring across 9 or more adjacent channels (minimum length 24 cm), where the waveforms in each channel overlapped with the adjacent channel, with a temporal offset. In addition, at least two pressure waves within the sequence had to have an amplitude ≥ 75 mmHg [6,11]. Once identified, it was then determined if they were associated with the following 2 motor patterns:
(i)*Pressurizations* (Figure 1C): as per our previous publication [6], we examined manometry traces for the presence of pressurizations that occurred at the end of HAPCs. Corsetti et al. [5] defined these as a simultaneous increase in pressure across all sensors of the manometry catheter distal to the conclusion of an HAPC;(ii)*Simultaneous pressure waves associated with HAPCs* (Figure 1D): Chen et al. [2] noted that HAPCs and SPWs could occur with the SPW passing through the HAPC. In this study, SPWs that passed through HAPCs were flagged, and we determined if these events were ‘colonic-only’ SPWs or ‘gastric/small bowel/colonic’ SPWs.

### 2.5. Symptoms

These studies formed part of a diagnostic investigation to assess the small bowel and colonic response to a range of introduced stimuli. The studies were not designed to examine the relationships that may exist between gastrointestinal motor patterns and symptoms. Furthermore, the young age of many of the subjects in this study (10 were ≤4 years) meant that an accurate account of abdominal symptoms was not possible. There were, however, 9 participants with an age range of 15–20 years. These patients noted abdominal symptoms such as pain, cramps, or feeling sick. For the younger children, the study investigators noted in a study diary if the child appeared unsettled or upset. The timing of major events, such as meals, drugs given, or a bowel motion, were all recorded. While every effort was made to keep the children lying still in bed during the recording, in the younger children, this was not always possible. 

### 2.6. Automated Identification of SPWs

The software written in Python (NumPy version 1.21.5) was used to identify SPWs. For this identification, the following steps were undertaken. 

The software first identified all time points in each recording where local pressure peaks occurred simultaneously across 3 or more channels. This allowed for the identification of times when simultaneous pressure waves could have occurred. These were initially labelled ‘candidate peak times’. Due to the noisiness of manometric data and the variability of noise across recordings, it was not practicable to formulate a hard threshold rule for the number of simultaneous channels that needed to be at peak pressure at a candidate peak time. Instead, the total number of local peaks occurring in any channel was calculated for each time point (red dots in Figure 2A). From this, we produced a *distribution* of peak counts across time. This distribution was smoothed, and its local maxima were identified as the candidate peak times (see bold red line at the top of Figure 2A).

The software also identified all elevated regions of pressure over the baseline for each channel (calculated as local mean over a 100 s moving window: blue regions in Figure 2B). A ‘candidate simultaneous event’ was formed by combining all elevated regions (at multiple adjacent sites) that overlapped a particular candidate peak time, resulting in a connected spatiotemporal region with a start and end time and start and end channel, in which pressure was simultaneously elevated in at least 80% of the channels and where at least 50% of the channels contained local pressure peaks aligned in time (peaks within the black rectangles in Figure 2C).

Candidate simultaneous events were discarded if they contained an insufficient number of *matching peaks*, i.e., local pressure peaks that occurred within 0.2 s of the candidate peak time. A candidate event needed to contain 8 matching peaks or else have matching peaks in at least 50% of its elevated regions. Candidate simultaneous events were also discarded if they contained too few elevated regions. A candidate needed to contain a minimum of 4 elevated regions and have elevated regions in 90% of its channels (Figure 2D).

The Identification of a set of candidate simultaneous events was carried out first within all recorded channels (stomach, small bowel, and colon), and then within the colonic channels only (blue regions in Figure 2E). Candidate simultaneous events were discarded from the colonic only set if they were also present in the whole recording set and extended into the small bowel and the stomach.

Candidate events that overlapped significantly in space and time were merged together. The software also searched for potential interrupted colonic pressurizations (see definitions above). When the software identified a sequence of several candidate simultaneous events occurring in rapid temporal succession and occurring at spatial locations that appeared to ‘advance’ aborally, they were merged into a single interrupted colonic pressurization.

Out of the resulting set of candidate simultaneous events, only those events that either spanned the whole gastric/small bowel and colon (‘gastric/small bowel/colonic’ SPW) recordings or events that spanned the entire colon only (‘colonic-only’ SPW) (Figure 2F) were retained.

Candidate events with a median amplitude below 2 mmHg were discarded as presumed respiratory or cardiac artefacts. The remaining simultaneous pressure waves formed the data set that was used in the data analysis as below.

All SPWs identified using this method were then cross-checked by a data analyst (RL) who manually examined the manometry trace in PlotHRM [12]. This allowed incorrectly labelled events to be discarded.

### 2.7. Parameter Extraction

For each SPW or interrupted colonic pressurization, the software calculated the amplitude and duration of the pressure waves. Amplitude was calculated as the median of peak amplitude across all elevated regions in an event, and duration was calculated as the median duration of all elevated regions. In the case of interrupted colonic pressurizations, this meant that amplitude and duration were calculated as medians across all regions in all segments that had been merged to form the event.

### 2.8. Statistical Analysis

Statistical analysis was performed using the Python module scipy.stats (version 1.5.0). Testing for significant differences between colonic only and gastric/small bowel/colonic events was performed using a paired t-test or Mann–Whitney U-test for parametric or nonparametric data, where appropriate. We also tested for significantly different ratios of events one hour before vs. one hour after the meal between colonic only and gastric/small bowel/colonic events using the chi-squared test with Yates’s correction. A *p* value of <0.05 was considered statistically significant.

Kernel density estimation on the amplitude and duration data was performed using the Python package seaborn (version 0.12.0).

## 3. Results

Of the 27 patients included in this analysis, 15 were female. The mean age was 9.3 + 1.2 years (range: 20 months–20 years). The patients had been referred for pan-intestinal manometry for the following reasons: 12 (43%) had severe feeding intolerance, 5 (18%) intractable nausea, 10 (36%) vomiting, 5 (18%) abdominal pain, 15 (54%) intractable constipation, and 1 (3.5%) a high stoma output. Eleven were receiving total parenteral nutrition at the time of investigation, and two were receiving jejunal feeding. Three had undergone an appendicostomy and three had an ileostomy. Intubation into the small bowel (pylorus to catheter tip) was 45 ± 12 cm (range: 32–68 cm). In the colon, the catheter tip was in the cecum in 13, hepatic flexure in 1, and proximal–mid-transverse colon in 11. In one patient, the catheter tip was located at the splenic flexure, and, in another, the catheter tip was in the sigmoid colon. As the sensors did not cover most of the colon in these two subjects, we did consider removing them from the analysis. However, as both had gastric/small bowel recordings, we were still able to identify all gastric/small bowel/colonic SPWs, and, therefore, they were left in the study. The mean manometry recording time from all 39 studies was 376 ± 110 min (range: 189 to 586 min). 

### 3.1. Identification of Simultaneous Pressure Waves (SPWs)

SPWs were found in each of the 39 recordings, with 14,565 identified in total (364 ± 316 SPWs/study; range: 5–1278). Of these, 14,550 (99.9%) spanned the entire antroduodenal (and, in some, oesophagal) and colonic recording sites and were, therefore, labelled as gastric/small bowel/colonic SPWs (Figure 3). Only *15* colonic only SPWs (0.1% of the total) were identified, all occurring in a single patient with constipation over an 80 min period (Figure 4). 

The SPWs spanning the entire stomach, small bowel, and colon occurred variably and irregularly over time, from 0 per minute to 40 per min (Figure 5). In most recordings, large numbers of SPW occurrences tended to be concentrated within specific periods of time, such as when the child was moving around, agitated, laughing, or crying (as taken from the study diaries). When the child was settled and lying still, there were relatively few events.

In addition to SPWs, there were 253 interrupted colonic pressurizations. These were identified in 10 of the 39 recordings (from eight different patients: Figure 4). In the one patient with colonic only SPWs, interrupted colonic pressurizations occurred amongst the colonic only SPWs at a rhythmic frequency of 1.3 cycles per min (Figure 4).

### 3.2. High Amplitude Propagating Contractions (HAPC)

HAPCs were identified in 21 of the 27 subjects. In these patients, 327 HAPCs were identified (mean 15.6 ± 11.6 HAPC; range 3–55), extending a mean distance of 40 ± 11 cm, propagating at a velocity of 10.9 ± 5.7 cm/s, with a mean amplitude of 124.0 ± 27.1 mmHg. The majority of HAPCs (219/327; 67%: n = 20 patients) were recorded after the intraluminal introduction of bisacodyl. Seventy-nine (24%) HAPCs were recorded during the basal period (n = 12 patients), and a further 29 (9%) were recorded after a meal (n = 7 patients).

Pressurizations were associated with 117 HAPCs (36%; Figure 6). Of these, 82 occurred after bisacodyl infusion, 27 occurred in association with HAPCs recorded during the basal period, and 10 occurred after a meal.

In addition to pressurizations, 188/327 HAPCs (57.4%) had SPWs pass through the propagating motor pattern (Figure 7). Every one of these SPWs also spanned all gastric and small bowel sensors (and the oesophagus, if sensors were present in that region: Figure 7).

### 3.3. Distinguishing Characteristics of Colonic Only SPW vs. Gastric/Small Bowel/Colonic SPW

As interrupted colonic pressurizations only occurred in the colon, the two colonic only motor patterns (colonic only SPWs and interrupted colonic pressurizations) were combined (yielding a combined set of 268 incidences) to compare their characteristics (amplitude and duration of pressure waves) against those of gastric/small bowel/colonic SPWs. The estimated probability density distribution of amplitude and duration values is shown in Figure 8A,B. Colonic only SPWs had a median peak amplitude (8.7 mmHg) that was significantly higher than gastric/small bowel/colonic SPWs (7.0 mmHg; *U* = 1,559,809, *p* < 0.001) and a median pressure-wave duration that was significantly longer (15.0 s) than gastric/small bowel/colonic SPWs (2.1 s; *U* = 117,354, *p* < 0.001). 

When attempting to predict for a given measurement of amplitude and duration, whether a particular SPW occurred in the colon only, or across the stomach, small bowel, and colon, the picture was less clear cut. For amplitude, the conditional probability of belonging to the gastric/small bowel/colonic SPW category was approximately 100% for all amplitude values, i.e., knowing the amplitude value offered no ability to predict the SPW category.

Attempts were made to characterize colonic only SPWs by the morphology of the pressure waves; however, given the wide range of pressure-wave morphologies in the gastric/small bowel/colonic SPWs, this was not possible.

The conditional probability of the SPW category (colonic only vs. gastric/small bowel/colonic), given the duration value, is shown in Figure 8C. While the probability of being a gastric/small bowel/colonic SPW was near 100% if the duration was <10 or >35 s, there was only a ~40–50% probability of being a colonic only SPW if the duration was between 15–25 s (Figure 8C).

### 3.4. Meal Response

An analysis was conducted on 25 recordings for which meal time information was available. When compared to the gastric/small bowel/colonic SPW events, those that occurred in the colon only (colonic only SPWs and interrupted colon pressurizations combined) were significantly more prevalent after the meal (χ^2^ with Yates’ correction = 11.120, *p* = 0.00085, df = 1). The meal had no effect upon the SPW that spanned the entire digestive tract.

### 3.5. Symptom and Associated Motor Patterns

Symptoms/comments were made in study diaries in 20 of the 39 recordings (n = 15 patients). The recorded symptoms were abdominal pain (n = 6), cramps (n = 3), nausea/vomiting (n = 8), gagging (n = 3), and bowel motion (n = 3). Additional comments were “appears unsettled” (n = 1), coughing (n = 2), yelling (n = 1), and crying (n = 4). On every occasion, each comment or symptom was preceded and/or followed by gastric/small bowel/colonic SPW. Examples can be seen in Figure 9. In one child, multiple examples of gastric/small bowel/colonic SPWs can be seen during a period in which the investigators noted the child appeared unsettled (Figure 9A). When the child was picked up and comforted by the mother, all SPWs stopped. In another child, abdominal cramps were assisted with a burst of colonic activity and a gastric/small bowel/colonic SPW was noted in the middle of the activity (Figure 9B). In a further child, a bowel action was preceded by multiple examples of HAPCs and gastric/small bowel/colonic SPWs (Figure 9C). A final example shows multiple gastric/small bowel/colonic SPWs associated with abdominal pain and crying (Figure 9D). In each of the examples shown in Figure 9, the recorded study diary comments, coupled with simultaneous activity spanning most of the digestive tract, indicate that SPWs are more likely to be caused by body movement, straining, or abdominal tightening rather than a motor pattern initiated as part of the normal colon physiology.

## 4. Discussion

In a pediatric patient population, the aim of this study was to characterize simultaneous pressure waves (SPWs) recorded via combined antroduodenal and colonic manometry and determine the proportion that occurred only in the colon versus those that spanned the entire digestive tract (gastric/small bowel/colonic). Overall, 14,565 SPWs spanned all recording sites in the colon. Of these, 14,550 (99.9%) also spanned all gastric/small bowel recording sites. There were only 15 SPWs that could be classified as ‘colonic-only’ SPWs, and these were all identified over an 80 min period in a single subject. In addition to SPWs, we identified 253 interrupted colonic pressurizations. These near simultaneous events were identified in eight patients. Further, high amplitude propagating sequences (HAPCs) were identified in 21 patients, of which around a third (36%) terminated in a simultaneous pressurization. Around half (57%) of the HAPCs had SPWs passing through the propagating motor pattern. However, all these SPWs also spanned the entire small bowel and gastric recording sites. Collectively, these findings suggest that the 99.9% of the SPWs recorded in this population are not specific colonic motor patterns and are more likely to be representative of extrinsic forces. Laughing, yelling, crying, sneezing, abdominal strain, and bending all cause body movement, and this movement would also cause either movement or compression of the digestive tract. Manometry sensors detect pressure and contact force, and, therefore, all of these factors could be detected as a simultaneous increase in pressure on a manometry trace.

A previous study by Corsetti et al. reported that events labelled as “pan-colonic pressurizations” commonly occurred in the healthy adult colon [5]. Their frequency increases after a meal [5] and, also, after prostigmine administration [13]. These motor patterns were also reported to be associated with anal sphincter relaxation [2,5]. Importantly, abdominal EMG recording showed that colonic pressurizations were *not* associated with any contraction of abdominal muscles and were, therefore, likely to be true colonic motor patterns. 

The Corsetti et al. study [5] also showed that pan-colonic pressurizations were significantly reduced in constipated patients. In the current study, 57% of the patients had constipation, which included the only patient who had colonic only SPWs. Previously, we failed to identify any colonic pressurizations in a study of 19 constipated children [14], supporting the concept that these motor events rarely exist in constipation. However, while colonic only SPWs were largely absent, there were eight children who displayed interrupted colonic pressurizations. These events consisted of multiple short extent simultaneous pressure waves occurring at spatial locations that appeared to propagate rapidly in an antegrade direction. In the one patient with colonic only SPWs, these interrupted colonic pressurizations occurred amongst the colonic only SPWs. Collectively, the two motor patterns occurred at a rhythmic interval of ~1.3 cycles per minute (see Figure 4). A similar frequency of colonic pressurizations had been noted by Corsetti et al. [5]. In our study, when the component pressure-wave duration of the interrupted colonic pressurizations and the colonic only SPWs were combined, an average pressure-wave duration of ~15 s was calculated. This duration lies within the mean ± 2SD of the duration of colonic pressurizations identified by Corsetti et al. [5]. Therefore, we suggest that the colonic only SPWs and the interrupted colonic pressurizations are likely to be the same type of motor pattern as the colonic pressurizations identified by Corsetti et al. [5].

In two other publications by Chen et al. [2,3], the pressure-wave duration of SPWs included a much broader range (2–57 s). The majority of SPWs recorded in the current study resembled SPWs in those studies (see Figure 6 in [3] and Figure 10 in [2]). An analysis of our data showed that 100% of SPWs with a duration of <10 s or >35 s spanned the entire digestive tract, from the stomach (or oesophagus) to the rectum. Given the extent of the regions involved and the differing neural innervation of those regions, it seems highly unlikely that these simultaneous pressure waves represent a highly coordinated pan-gut contraction. 

Ten of the patients involved in the current study were young children (<4 years old), and episodes of crying, yelling and general movement were not uncommon. When such episodes were detailed in the study diary, they were always temporally associated with multiple gastric/small bowel/colonic SPWs. Such patterns would be expected, as body motion and abdominal strain have long been associated with a simultaneous increase in pressure across all manometry sensors. While gastric/small bowel/colonic SPWs may be expected in young children, they were also just as prolific in the nine patients aged between 15 and 20, indicating that simultaneous pressure waves are not just a phenomenon occurring in young children. Across the entire age spectrum, gastric/small bowel/colonic SPWs occurred at multiple different frequencies, and, unlike colonic only events, their numbers were not influenced by a meal. This also indicates that gastric/small bowel/colonic SPWs are not influenced by stimuli known to influence colonic motility. In addition, a similar motor to SPWs has been described in patients with rumination using antroduodenal manometry [15,16]. In those studies, the simultaneous pressure increases were associated with EMG changes, indicating the pressure waves were generated extrinsically.

SPWs have also been shown to occur with HAPCs (i.e., passing through the propagating motor event). It has been proposed that the presence of HAPCs is an indicator of normal colonic musculature and enteric neural circuits [17]. SPWs occurring concomitantly with HAPCs have been proposed as further evidence of extrinsic neural reflexes [2]. In the current study, most subjects exhibited HAPCs following bisacodyl administration or during the basal/meal recordings. Just over half of these HAPCs had SPWs passing through them. These motor patterns look equivalent to those shown by Chen et al. (see Figure 13 in [2]). However, in our patients, all of the HAPC-associated SPWs were shown to span the majority of the gut, including the oesophagus if sensors were present in that region. HAPCs have been associated with defecation and an urge to defecate [18,19], both of which may be associated with straining, tightening of the abdominal cavity, or general body movement (shifting to get more comfortable waiting for the urge to pass). In the current patient population, these factors external to the colon would seem to provide a plausible explanation of the occurrence of SPW during an HAPC. 

There are several key limitations to this study. Due to a lack of maintenance of the sensor(s) within the anal canal allied to sensor-spacing employed, we were unable to comment upon any association between SPWs and anal sphincter relaxation. In addition, while previous studies have associated SPWs with flatus and possibly gas transit [2,5], there was no marking of flatus in this study. However, given the high number of SPWs identified (364 ± 316 SPW/study), and the fact that the daily frequency of flatus ranges from 10 to 19 episodes [20,21], it is implausible that SPWs in this study were associated with flatus. Given that we did not measure EMG and all body movements of the patients during the recordings, we cannot say with certainty that the gastric/small bowel/colonic SPWs were all associated with body movement or straining. However, when periods of crying, yelling, or unsettled behaviour were documented, they were *always* associated with multiple SPWs. To confirm a genuine association between the artefact and gastric/small bowel/colonic SPWs, abdominal EMG recordings would be needed [5].

## 5. Conclusions

The current data show that virtually all (99.9%) SPWs recorded during colonic manometry studies are also simultaneously recorded at sites located in the stomach and proximal small bowel (and the oesophagus if sensors were present in the region). For SPWs with a pressure-wave duration of <10 s or >35 s, 100% were simultaneously recorded in the colon and upper GI tract. Colonic only SPWs were identified in one patient. Based on these findings, we suggest that, in pediatric studies, SPWs should not form part of any diagnostic criteria, as most events appear to be an artefact caused by factors outside the colon (abdominal strain and/or body motion). The exceptions to this are pressurizations at the end of an HAPC and interrupted colonic pressurizations, both of which were only seen in the colon.

## Figures and Tables

**Figure 1 jcm-12-05808-f001:**
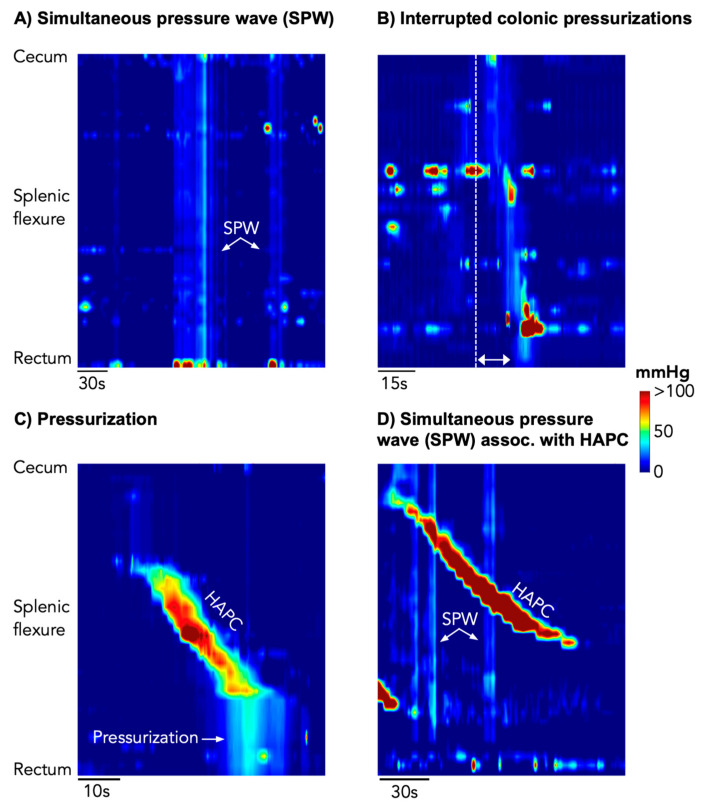
Types of simultaneous pressure waves identified.

**Figure 2 jcm-12-05808-f002:**
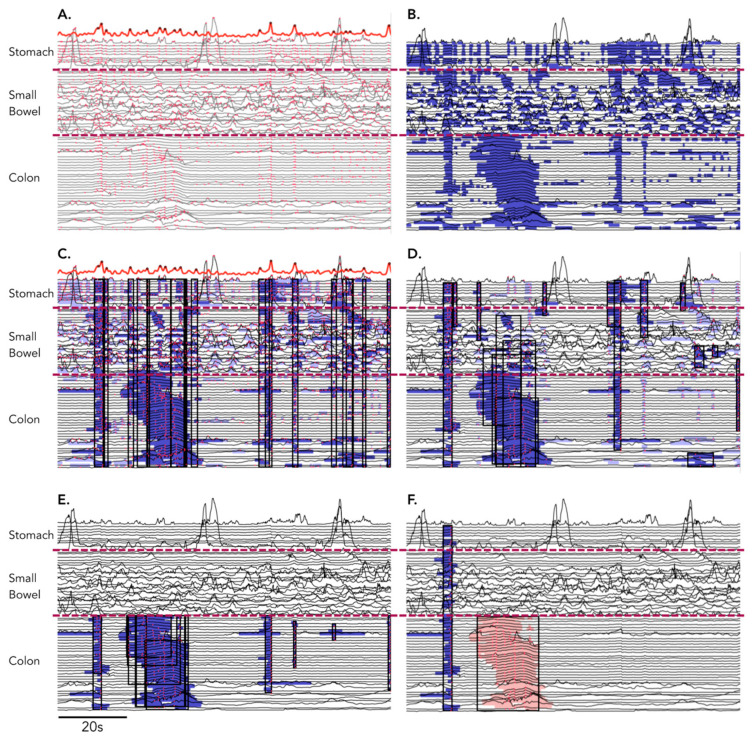
Automated identification of simultaneous pressure waves. The steps (**A**–**F**) are detailed in the methodology section. The red hatched lines represent the anatomical division between the stomach, small bowel and colon. The solid red line in (**A**,**C**) represents the smoothed distribution of peaks located in the manometry trace (see Automated Identification of SPWs for details). The shaded blue regions represent candidate potential simultaneous events. The shaded red region in (**F**) represents a colon only event.

**Figure 3 jcm-12-05808-f003:**
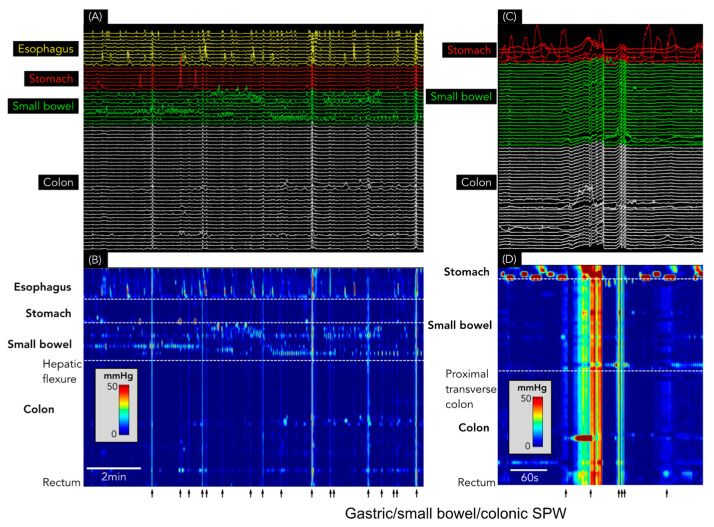
(**A**) Conventional line plot of simultaneous pressure waves spanning the oesophagus (yellow), stomach (red), small bowel (green), and colon (white). (**B**) shows the same data as a spatiotemporal colour map. Note that the simultaneous pressure waves (identified by each black arrow) span the entire digestive tract from the oesophagus to the rectum. (**C**) Conventional line plot shows the gastric/small bowel/colonic SPW. (**D**) shows the same data as a spatiotemporal colour map. In these examples, the SPW can be seen to consist of varying durations, from <5 s to >30 s.

**Figure 4 jcm-12-05808-f004:**
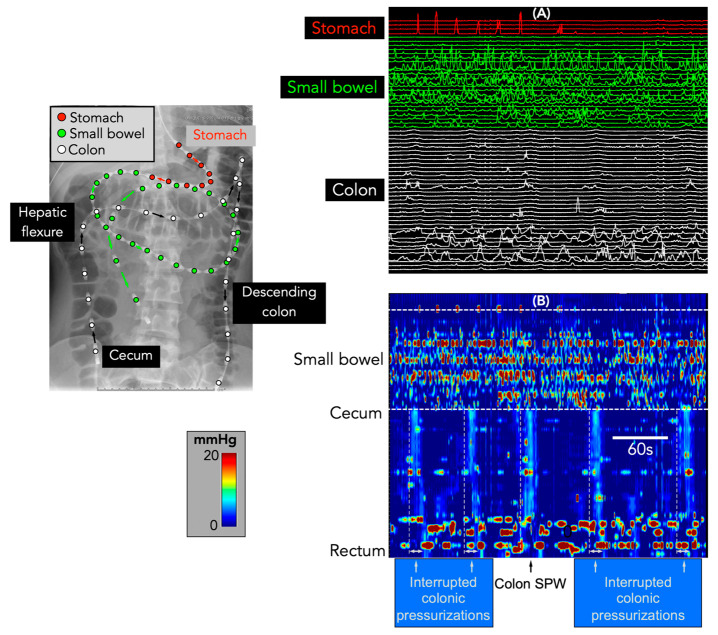
Interrupted colonic pressurizations and colonic only-SPW. The X-ray image shows the placement of the catheter in the colon (white circles), small bowel (green circles), and stomach (red circles). (**A**) shows the line plot of the manometry from the 3 regions shown in the X-ray. (**B**) shows the same data shown as a spatiotemporal colour map. The colonic only-SPW appears as a simultaneous increase in pressure along all sensors in the colon. The simultaneous pressure waves were identified in just 1 patient. The interrupted colonic pressurizations show a clear time lag between the upstroke of the pressure waves in the proximal compared to the distal colon. Note that both events span the entire length of the colon, and the pressure increases are not seen in the small bowel or stomach.

**Figure 5 jcm-12-05808-f005:**
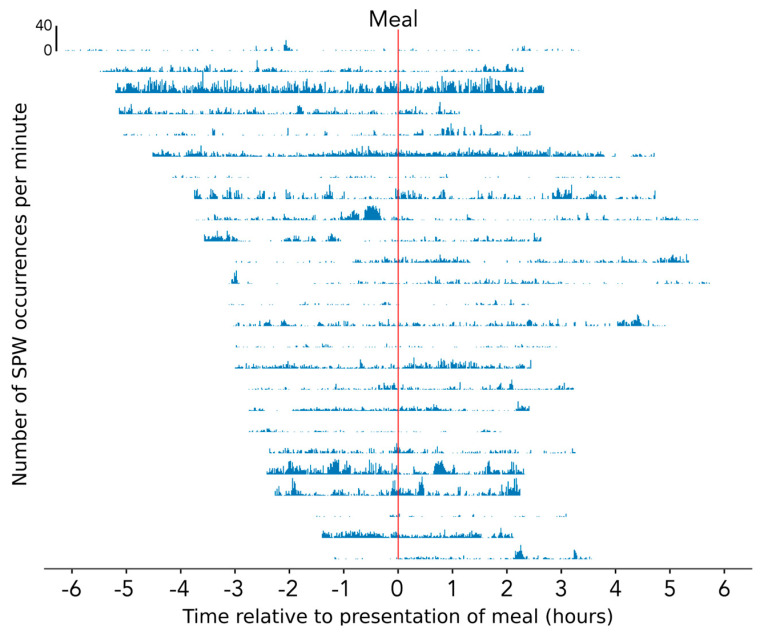
Histograms showing the time distribution of gastric/small bowel/colonic SPW. These figures show the data from subjects that had a meal (n = 25). Each individual row shows the frequency of occurrence of the SPWs in an individual patient over the course of a single recording with time indicated relative to the meal (vertical red line). Each individual bar in the histogram represents the number of SPWs observed within one minute, ranging between zero and a maximum observed frequency of 40 per minute.

**Figure 6 jcm-12-05808-f006:**
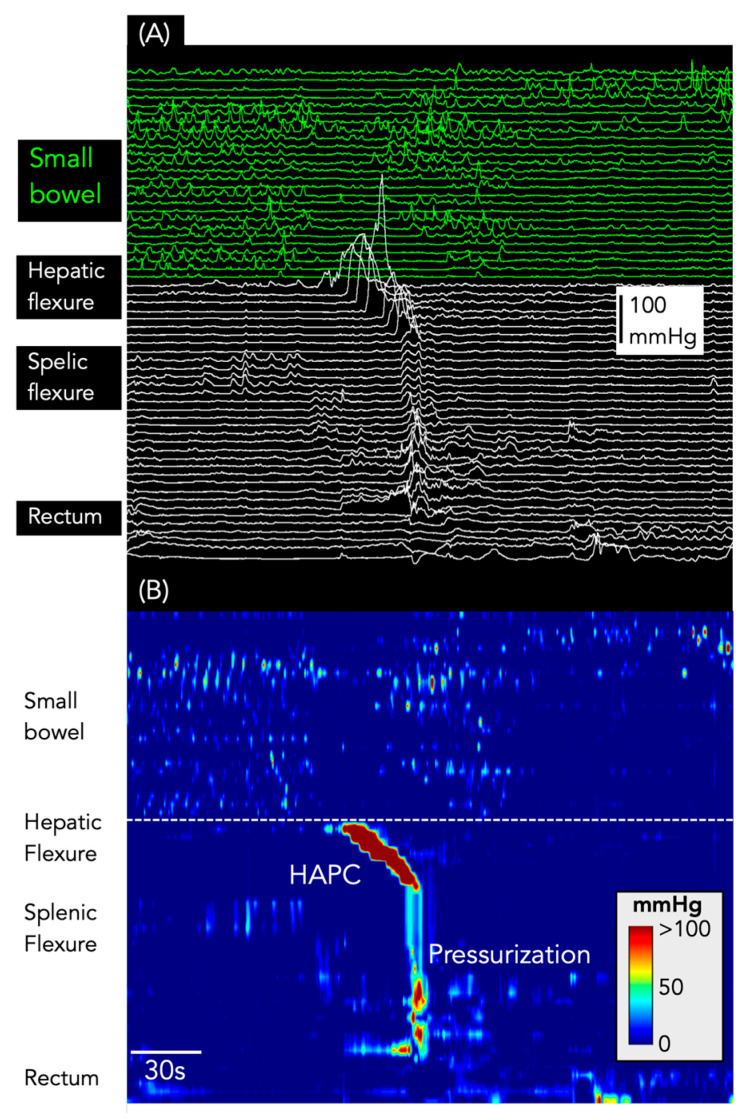
Pressurizations following a HAPC. (**A**) shows the line plot of the manometry from the small bowel (green) and colon (white). (**B**) shows the same data shown as a spatiotemporal colour map.

**Figure 7 jcm-12-05808-f007:**
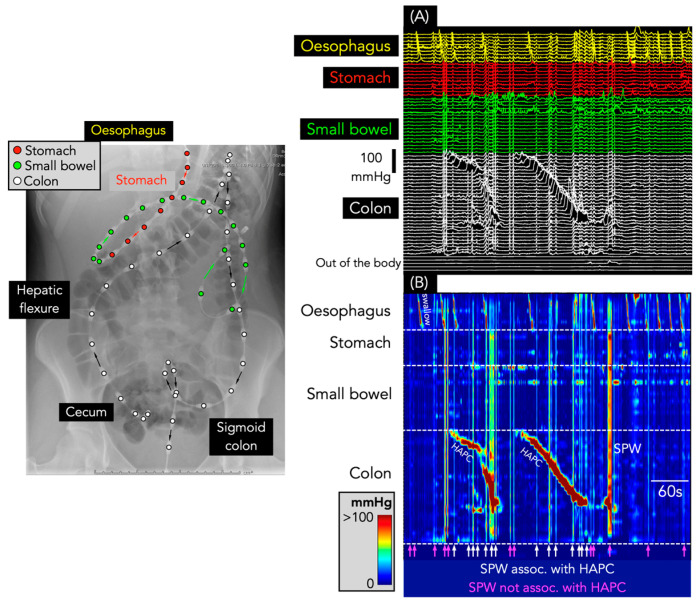
Simultaneous pressure waves associated with HAPCs. The X-ray image shows the placement of the catheter in the colon (white circles), small bowel (green circles), and stomach (red circles). The oesophageal sensors are not shown in the X-ray. (**A**) shows the line plot of the manometry. (**B**) shows the same data shown as a spatiotemporal colour map. Note that all SPWs span the stomach, small bowel, and colon and, in most instances, they are also detected by the sensors in the oesophagus. All of the SPWs are likely to represent body motion or abdominal strain (artefact) rather than genuine gut contractions.

**Figure 8 jcm-12-05808-f008:**
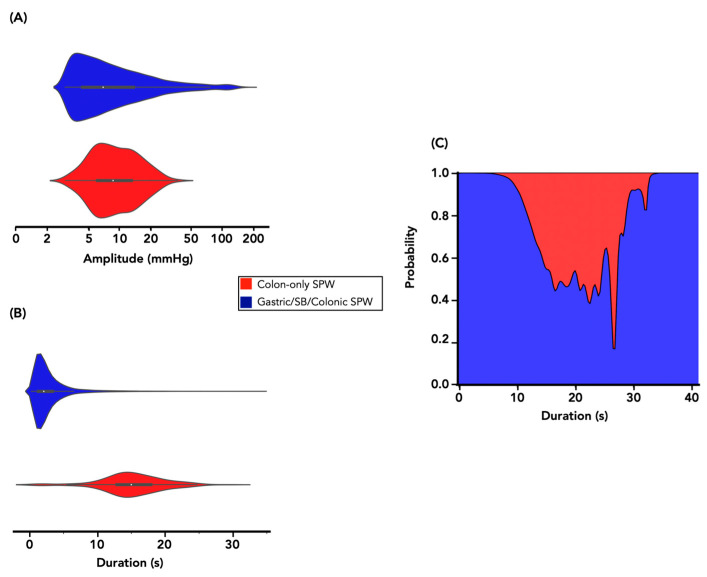
Kernel estimation of the probability density of (**A**) peak amplitude and (**B**) the duration of simultaneous pressure waves in this study, separately for colonic only SPWs (red) and gastric/small bowel/colonic SPWs (blue). (**C**) The estimated conditional probability, given event duration, of simultaneous pressure waves in this study being in the stomach, small bowel, and colon (blue) versus in the colon only (red).

**Figure 9 jcm-12-05808-f009:**
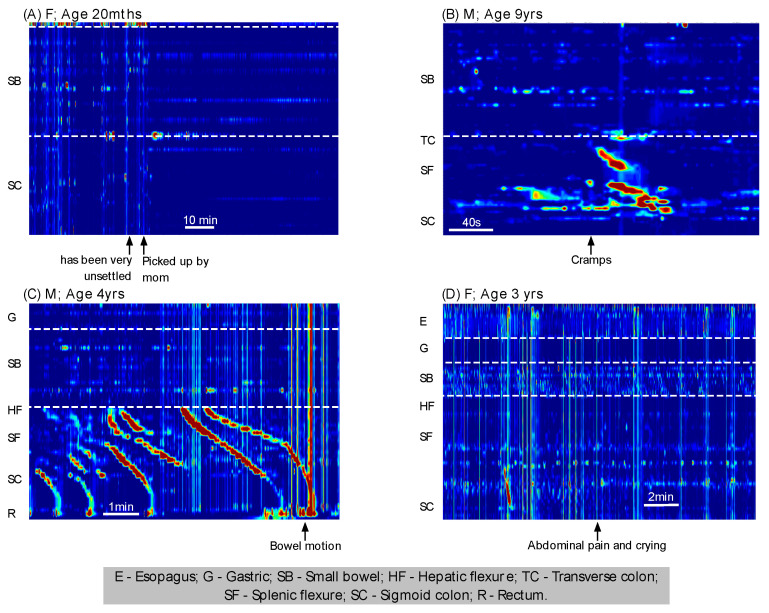
Gastric/small bowel/colonic SPWs associated with comments made on the manometry trace. These motor patterns were observed in association with being (**A**) unsettled, (**B**) abdominal cramps, (**C**) a bowel motion, and (**D**) abdominal pain and crying.

## Data Availability

Data can be made available upon request with the appropriate data-sharing agreements in place.

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
