# Peer review of "Simultaneous Colonic Pressure Waves in Children and Young Adults with Gastrointestinal Motility Disorders: Artefact or Colonic Physiology?"

_jcm, 2023, doi:10.3390/jcm12185808_

Round 1

Reviewer 1 Report

The authors describe that SPWs appear to be artifacts caused by factors outside the colon. In an adult study, SPWs were measured by colorectal HRM, and their association with constipation was pointed out. As the authors mentioned, most of the SPWs were measured in the entire gastrointestinal tract using both gastroduodenal and colonic HRM. So, it’s important to show that SPWs should not form part of any diagnostic criteria for the world.

However, some revisions are necessary as the following points. I would like for the author to answer the following questions.

 1.     In the methods (line 89), the authors described the colonic manometry catheter was placed with the use of a colonoscope and guide wire as previously described (9). In reference no. 9, the authors described that the CM was performed as per the previously described protocol. So, please add the original papers on colonic manometry catheter placement to reference.

 2.     What do the authors think about the mechanism of SPWs? If laughing or crying increases abdominal pressure, does it also increase intra-intestinal pressure?

 3.     It seems that conditions such as physical activity are different between adults and children in HRM studies. Do you have experience with HRM comparing adults and children?

Author Response

REVIEWER 1

Comments and Suggestions for Authors

The authors describe that SPWs appear to be artifacts caused by factors outside the colon. In an adult study, SPWs were measured by colorectal HRM, and their association with constipation was pointed out. As the authors mentioned, most of the SPWs were measured in the entire gastrointestinal tract using both gastroduodenal and colonic HRM. So, it’s important to show that SPWs should not form part of any diagnostic criteria for the world.

However, some revisions are necessary as the following points. I would like for the author to answer the following questions.

  1. In the methods (line 89), the authors described the colonic manometry catheter was placed with the use of a colonoscope and guide wire as previously described (9). In reference no. 9, the authors described that the CM was performed as per the previously described protocol. So, please add the original papers on colonic manometry catheter placement to reference.

Thankyou for highlighting this. The correct reference should have been “Rodriguez L, Sood M, Di Lorenzo C, Saps M. An ANMS-NASPGHAN consensus document on anorectal and colonic manometry in children. Neurogastroenterol Motil. 2017 Jan;29(1). doi: 10.1111/nmo.12944”.

This has been corrected in the paper.

  1. What do the authors think about the mechanism of SPWs? If laughing or crying increases abdominal pressure, does it also increase intra-intestinal pressure?

The sensors in these catheters respond to contact force and changes in intra luminal pressure. Body movement caused by laughing etc., would cause simultaneous movement of the entire body including the digestive tract, and this would be detected by the catheters. Abdominal strain and bending over is likely to cause an increase in intestinal pressure because such movements would compress the abdominal cavity. We have altered the end of the first paragraph in the discussion to reflect these comments (Lines 409-413)

  1. It seems that conditions such as physical activity are different between adults and children in HRM studies. Do you have experience with HRM comparing adults and children?

Paediatric studies are more of a challenge; it is much harder to keep children still and get them to eat standardised meals. One of the main problems we have with paediatric studies is that there is no healthy control group to use as a comparator. As such we always use our healthy adult studies as a control group . Therefore, we have had experience in comparing adult and children colonic manometry studies (1-4). As yet we have not identified any age specific genuine colonic motor patterns.

References

  1. King SK, Catto-Smith AG, Stanton MP, et al. 24-Hour colonic manometry in pediatric slow transit constipation shows significant reductions in antegrade propagation. Am J Gastroenterol 2008;103:2083-91.
  2. Stanton MP, Hutson JM, Simpson D, et al. Colonic manometry via appendicostomy shows reduced frequency, amplitude, and length of propagating sequences in children with slow-transit constipation. J Pediatr Surg 2005;40:1138-45.
  3. Koppen IJN, Wiklendt L, Yacob D, Di Lorenzo C, Benninga MA, Dinning PG. Motility of the left colon in children and adolescents with functional constpation; a retrospective comparison between solid-state and water-perfused colonic manometry. Neurogastroenterol Motil 2018;30:e13401.
  4. Wessel S, Koppen IJ, Wiklendt L, Costa M, Benninga MA, Dinning PG. Characterizing colonic motility in children with chronic intractable constipation: a look beyond high-amplitude propagating sequences. Neurogastroenterol Motil 2016;28:743-57.

Reviewer 2 Report

1. Please improvise your introduction. Add more about clinical relevance and if data are available, provide some information on the current prevalence and incidence.

Research is interesting and overall quality is very nice.

Author Response

REVIEWER 2

Comments and Suggestions for Authors

Please improvise your introduction. Add more about clinical relevance and if data are available, provide some information on the current prevalence and incidence.

Research is interesting and overall quality is very nice.

We are not sure what the reviewer means by “improvise your introduction”, but as the introduction currently stands, we have already stated the potential clinical relevance of simultaneous pressure waves on lines 44-48. The reviewer also asked to provide “some information on the current prevalence and incidence”, this has been added at lines 40-41.

Reviewer 3 Report

I would like to congratulate the authors for presenting high-quality original research on an important subject. The present work investigates the relevance of “pancolonic” simultaneous pressure waves in high resolution colonic manometry studies in children. Certainly, in motility tracings, simultaneous increases in pressure across long segments of the gut must raise suspicions for artifacts. The longer the segment, the higher the probability of the event being external to the gut lumen. To do so, the authors retrospectively review data from children with clinical indications, that were evaluated by pan enteric simultaneous manometry. Indeed, the vast majority of simultaneous colonic pressure waves in these tracings were present along all sensors, and thus should be considered artifacts. The manuscript is clear and well written, the research mythology is described in detail and is appropriate to reach this conclusion.

My only concern with the manuscript is the comparison between the author’s results with the studies performed by Corsetti et al. and Chen et al., a comparison that might be problematic. Mainly because the simultaneous colonic pressure waves were described in healthy asymptomatic adults that volunteered to participate in the study; these volunteers seemingly were highly motivated and cooperative during the study, the appearance of artifacts in these kinds of studies is expected to be low. Indeed, Corsetti et. al. went a long way to very that the waves that they observed were not artifacts. A very different situation from a child or an adolescent being studied for refractory gut symptoms, a situation in which the frequency of artifacts is expected to be high.  I suggest you modify the discussion to reflect these differences.

Nevertheless, I agree that the findings in this study call for a replication study in adults.  

Author Response

REVIEWER 3

Comments and Suggestions for Authors

I would like to congratulate the authors for presenting high-quality original research on an important subject. The present work investigates the relevance of “pancolonic” simultaneous pressure waves in high resolution colonic manometry studies in children. Certainly, in motility tracings, simultaneous increases in pressure across long segments of the gut must raise suspicions for artifacts. The longer the segment, the higher the probability of the event being external to the gut lumen. To do so, the authors retrospectively review data from children with clinical indications, that were evaluated by pan enteric simultaneous manometry. Indeed, the vast majority of simultaneous colonic pressure waves in these tracings were present along all sensors, and thus should be considered artifacts. The manuscript is clear and well written, the research mythology is described in detail and is appropriate to reach this conclusion.

My only concern with the manuscript is the comparison between the author’s results with the studies performed by Corsetti et al. and Chen et al., a comparison that might be problematic. Mainly because the simultaneous colonic pressure waves were described in healthy asymptomatic adults that volunteered to participate in the study; these volunteers seemingly were highly motivated and cooperative during the study, the appearance of artifacts in these kinds of studies is expected to be low. Indeed, Corsetti et. al. went a long way to very that the waves that they observed were not artifacts. A very different situation from a child or an adolescent being studied for refractory gut symptoms, a situation in which the frequency of artifacts is expected to be high.  I suggest you modify the discussion to reflect these differences. Nevertheless, I agree that the findings in this study call for a replication study in adults.  

The thank the reviewer for the positive feedback and we also agree that a direct comparison between the work by Corsetti and our data is problematic. Indeed, this is always a problem in paediatric manometry studies; there are never any directly comparable healthy control studies (you cannot do manometry in a healthy child). Nevertheless, more colonic manometry studies are done in children/teenagers than in adults and in children the procedure is used as a diagnostic tool. Therefore, we must determine if motor patterns identified in healthy adults are also found in children. This becomes even more essential when authors describe a motor pattern as having potential diagnostic value (1).

The Corsetti study provides data on the duration and amplitude of the SPW, and we have already used these criteria in one of our previous published paediatric studies (2) (mentioned on lines 423-433 of the paper). As we can identify all other described adult motor patterns in children (3-5) we are confident that if the motor pattern exists then it can identified even if there is a lot of body movement. Our data did show that one child had a motor pattern similar to those described by Corsetti. However, importantly, our data also show that SPWs should not be used as part of any diagnostic criteria in children, because nearly all are clearly artefact. Finally in the limitations section of the discussion we do clearly state the clear differences in recording between our study and the Corsetti paper.

References

  1. Chen J-H, Parsons SP, Shokrollahi M, et al. Characterization of simultaneous pressure waves as biomarkers for colonic motility assessed by high-resolution colonic manometry. Frontiers in physiology 2018;9:1248.
  2. Koppen IJN, Wiklendt L, Yacob D, Di Lorenzo C, Benninga MA, Dinning PG. Motility of the left colon in children and adolescents with functional constpation; a retrospective comparison between solid-state and water-perfused colonic manometry. Neurogastroenterol Motil 2018;30:e13401.
  3. King SK, Catto-Smith AG, Stanton MP, et al. 24-Hour colonic manometry in pediatric slow transit constipation shows significant reductions in antegrade propagation. Am J Gastroenterol 2008;103:2083-91.
  4. Stanton MP, Hutson JM, Simpson D, et al. Colonic manometry via appendicostomy shows reduced frequency, amplitude, and length of propagating sequences in children with slow-transit constipation. J Pediatr Surg 2005;40:1138-45.
  5. Wessel S, Koppen IJ, Wiklendt L, Costa M, Benninga MA, Dinning PG. Characterizing colonic motility in children with chronic intractable constipation: a look beyond high-amplitude propagating sequences. Neurogastroenterol Motil 2016;28:743-57.